# A Fermented Milk Product with *B. lactis* CNCM I-2494 and Lactic Acid Bacteria Improves Gastrointestinal Comfort in Response to a Challenge Diet Rich in Fermentable Residues in Healthy Subjects

**DOI:** 10.3390/nu12020320

**Published:** 2020-01-25

**Authors:** Boris Le Nevé, Adrian Martinez-De la Torre, Julien Tap, Muriel Derrien, Aurélie Cotillard, Elizabeth Barba, Marianela Mego, Adoración Nieto Ruiz, Laura Hernandez-Palet, Quentin Dornic, Jean-Michel Faurie, John Butler, Xavi Merino, Beatriz Lobo, Ferran Pinsach Batet, Anna Accarino, Marta Pozuelo, Chaysavanh Manichanh, Fernando Azpiroz

**Affiliations:** 1Danone Nutricia Research, 91767 Palaiseau, France; Julien.TAP@danone.com (J.T.); Muriel.DERRIEN@danone.com (M.D.); aurelie.cotillard@danone.com (A.C.); quentin.dornic@orange.fr (Q.D.); Jean-Michel.FAURIE@danone.com (J.-M.F.); 2Digestive System Research Unit, University Hospital Vall d’Hebron; Centro de Investigación Biomédica en Red de Enfermedades Hepáticas y Digestivas (Ciberehd); Departament de Medicina, Universitat Autònoma de Barcelona, 08193 Cerdanyola del Vallès, Spain; adriadelatorre@hotmail.com (A.M.-D.l.T.); ebarbaorozco@gmail.com (E.B.); marianelamego@hotmail.com (M.M.); anieto@vhebron.net (A.N.R.); lhernandezpalet@gmail.com (L.H.-P.); xavier.merino@uab.cat (X.M.); beatriz.lobo@vhir.org (B.L.); f.pinsach.batet@gmail.com (F.P.B.); aaccarino@vhebron.net (A.A.); mpozud00@gmail.com (M.P.); cmanicha@gmail.com (C.M.); 3Lawson Health Research Institute, London, ON N6C 2R5, Canada; jbutler@lawsonimaging.ca

**Keywords:** flatulence, fermentable carbohydrates, probiotics, microbiota, digestive symptoms

## Abstract

Background: Healthy plant-based diets rich in fermentable residues may induce gas-related symptoms. Our aim was to determine the potential of a fermented milk product with probiotics in improving digestive comfort with such diets. Methods: In an open design, a 3-day high-residue diet was administered to healthy subjects (*n* = 74 included, *n* = 63 completed) before and following 28 days consumption of a fermented milk product (FMP) containing *Bifidobacterium animalis* subsp. *lactis* CNCM I-2494 and lactic acid bacteria. Main outcomes: digestive sensations, number of daytime anal gas evacuations, and gas volume evacuated during 4 h after a probe meal. Results: As compared to the habitual diet, the high-residue diet induced gas-related symptoms (flatulence score 4.9 vs. 1.2; *p* ≤ 0.0001), increased the daily number of anal gas evacuations (20.7 vs. 8.7; *p* < 0.0001), and impaired digestive well-being (1.0 vs. 3.4; *p* < 0.05). FMP consumption reduced flatulence sensation (by −1.7 [−1.9; −1.6]; *p* < 0.0001), reduced the number of daily evacuations (by −5.8 [−6.5; −5.1]; *p* < 0.0001), and improved digestive well-being (by +0.6 [+0.4; +0.7]; *p* < 0.05). FMP consumption did not affect the gas volume evacuated after a probe meal. Conclusion: In healthy subjects, consumption of a FMP containing *B. lactis* CNCM I-2494 and lactic acid bacteria improves the tolerance of a flatulogenic diet by subjective and objective criteria (sensations and number of anal gas evacuations, respectively).

## 1. Introduction

A substantial proportion of patients in the gastroenterology clinic relate their symptoms to intestinal gas. Typically bloating, abdominal distension, and flatulence are attributed to intestinal gas. Studies showed that 15%–20% of the general population in US and Europe experience these symptoms [1,2] and the proportion increases up to 90% in patients with functional digestive symptoms, i.e., without detectable abnormalities by conventional testing, particularly in patients with irritable bowel syndrome (IBS) [3]. Since the cause of these symptoms is uncertain, the current treatments are of limited efficacy [4].

Most of the gas in the digestive tract is produced in the colon by intestinal microbiota in the process of fermentation of meal residues that escape small bowel absorption [5]. Particularly, it has been shown that some food components, such as resistant starches, cellulose, or pectins, are incompletely absorbed in the small bowel and enter the colon [6,7,8]. The physiological role of the different gases and volatile compounds produced by colonic microbiota is incompletely understood, but some products of fermentation play a positive role as anti-inflammatory, anti-oxidative, and neuroprotective agents.

Within subjects, the volume of gas output varies in relation to the diet [9]. However, there is a great inter-individual variability, and gas evacuation in subjects maintained on a similar diet may differ substantially. This depends mainly on the composition and metabolic activity of the individuals’ colonic microbiota [5]. Hence, the volume of gas production and anal evacuation is determined by two main factors: the diet, particularly the amount of fermentable residues reaching the colon, and the individual composition and activity of the colonic microbiota. Indeed, human gut microbiota encodes for a large variety of carbohydrate-active enzymes [10] and this may vary between subjects.

A previous study [9] showed that a diet rich in fermentable residues significantly increased the volume and the number of anal gas evacuations, and induced gas-related symptoms in healthy subjects. This challenge diet included legumes among other high-residue foodstuffs. Legumes contain high levels of indigestible alpha-galactosides that are exclusively fermented by resident microbiota, releasing gas [11].

A fermented milk product containing *Bifidobacterium animalis* subsp. *lactis* CNCM I-2494 and lactic acid bacteria has been shown to improve symptoms and well-being in women with mild digestive complaints [12,13,14,15], and to improve bloating, digestive discomfort, and reduce abdominal distension in IBS-C patients [16,17]. The response to this product seems to be related to the metabolic activity of the gut microbiota [18].

In the present study, we hypothesized that a fermented milk product containing the same *B. lactis* CNCM I-2494 and lactic acid bacteria may reduce subjective and objective components of flatulence in healthy subjects challenged by a diet rich in fermentable residues.

## 2. Materials and Methods

### 2.1. Study Subjects

Healthy subjects (both genders; 18–75 years age range; body mass index, BMI. 18.5–30 Kg/m^2^) without gastrointestinal symptoms or history of gastrointestinal disorders, including food allergies and intolerances, and following a non-restrictive, omnivorous diet participated in the study. Participants were given standardized instructions to carefully fill out a clinical questionnaire based on Rome III criteria in front of the investigator. Participants were required to have normal bowel habits and no functional gastrointestinal disorder (no symptom ≥ 2 on a 0–10 scale). The standardization quality and sensitivity of this questionnaire have been validated by previous studies showing that it allows a good discrimination between patients and healthy subjects [9,19,20,21]. Exclusion criteria included any change in dietary habits in the previous 4 weeks, intake of antibiotics during the previous two months, treatments that might modify gastrointestinal function and/or affect the central nervous system (opioids, antidiarrheal agents, prokinetics, spasmolytics, laxatives), and any antecedents of digestive surgery (except for appendectomy and cholecystectomy performed more than two years before). Participants were recruited by public advertisement and received a monetary compensation for their participation in the study. The food products of the challenge diet and the fermented milk product were provided by the investigators. The protocol was approved by the Institutional Review Board of University Hospital Vall d’Hebron and all participants gave written informed consent prior to inclusion.

### 2.2. Study Design

This single center open label study was performed between October 2014 and December 2016 in a tertiary care referral center. The study protocol was registered with ClinicalTrials.gov (NCT02936713). The registered protocol included separate pilot studies in healthy subjects (reported here) and in patients with functional digestive disorders. All authors had access to the study data and reviewed and approved the final manuscript. The study included an 18-day run-in phase and a 28-day administration phase (Figure 1). The main outcomes were the differences in gastrointestinal symptoms, number of gas evacuations, and volume of gas evacuated after a probe meal measured at the end of the run-in phase and the administration phase. A 15-day period in the run-in phase was allowed before the flatulogenic diet to avoid potential carry-over effects of probiotic consumption in the habitual diet of the participants prior to their inclusion in the study.

During the study, participants consumed their habitual diet except during the last 3 days of the run-in phase (days 16–18) and the administration phase (days 44–46) when a flatulogenic diet (see below) was administered. The day after the run-in phase (day 19) subjects were re-evaluated, and only subjects who fulfilled the 2 following continuation criteria entered the administration phase: (a) ≥50% daily compliance to the flatulogenic diet (calculated as the percent intake per day of the total fiber content in the diet); and (b) an increase in flatulence score ≥2 on a 0–10 scale during the flatulogenic diet as compared to the habitual diet. For the duration of the study, subjects were not allowed to consume any fermented dairy products (with or without probiotics) or any tablets, pills, or food supplements containing pre- or probiotics other than those provided.

### 2.3. Challenge Diet

The challenge diet [22] consisted of: (a) breakfast of wholemeal cookies (39 g) plus coffee, tea, and/or milk; (b) lunch of white beans (200 g), mixed vegetables (250 g) or chickpeas (200 g) and wholemeal bread (50 g), plus meat, fowl or fish, and fruit (banana, figs, peaches or prunes); and (c) dinner of vegetable soup (200 mL), wholemeal bread (50 g) and fruit (banana, figs, peaches, or prunes). This diet provides 61% caloric content as carbohydrates, 25% proteins and 14% fat with 27 g fiber per day. The caloric content of the diet was not standardized. Participants were instructed to fill out a diary specifying the foods they consumed during the 3 days on the flatulogenic diet to assess compliance. This challenge diet increases intestinal gas production and induces symptoms [22].

### 2.4. Study Product

During the 28-day administration phase, subjects consumed 1 pot (125 g) of the study product at breakfast and 1 pot at dinner. The study product was a fermented milk containing three *Streptococcus salivarius* subsp. *thermophilus* strains (CNCM I-2773, CNCM I-2130, CNCM I-2272), *Lactobacillus delbrueckii* subsp. *bulgaricus* (CNCM I-1519), *Bifidobacterium animalis* subsp. *lactis* (CNCM I-2494), and *Lactococcus lactis* subsp. *lactis* (CNCM I-1631). The study product was manufactured and supplied by Danone Nutricia Research, Palaiseau, France and contained per g at least 3.4 × 10^7^ colony forming units (cfu) of *B. lactis*, 1 × 10^6^ cfu of *L*. *lactis*, and 1 × 10^7^ cfu of *S. thermophilus* and *L. bulgaricus*.

### 2.5. Main Outcomes

Digestive sensations and the number of anal gas evacuations were measured during 3-day periods at 3 time points throughout the study: (a) at the beginning of the run-in phase on the habitual diet (days 1–3); (b) at the end of the run-in phase on the challenge diet (days 16–18); and (c) at the end of the administration phase on the challenge diet (days 44–46) (Figure 1). The volume of gas evacuated after a probe meal was measured after the 3 days challenge diet on 2 occasions: (a) the day after the run-in phase (day 19); and (b) the day after the administration phase (day 47).

#### 2.5.1. Daily Symptoms Questionnaire

During the 3 days of each evaluation period, digestive sensations were measured using daily questionnaires that included 0–10 analogue scales for scoring: (a) subjective sensation of flatulence (defined as anal gas evacuation); (b) abdominal bloating (pressure/fullness); (c) abdominal distension (sensation of girth increase); (d) borborygmi; (e) odoriferous flatus; and (f) abdominal discomfort/pain; the questionnaire also recorded: (g) digestive well-being on a scale graded from +5 (extremely pleasant sensation/satisfaction) to −5 (extremely unpleasant sensation/dissatisfaction); (h) number of bowel movements; and (i) stool form using the Bristol stool form scale. Participants were given standard instructions to fill out scales by the end of the day. The sensitivity and quality of this questionnaire have been validated by previous studies showing the detection of the effects of dietary interventions in populations of healthy subjects and patients with functional gut disorders [9].

#### 2.5.2. Number of Anal Gas Evacuations

The number of daytime anal gas evacuations was measured during each evaluation period. Participants were instructed to carry an event marker (Hand Tally Counter No 101, Digi Sport Instruments, Shangqiu, China) during the day and to use it to register each passage of anal gas.

#### 2.5.3. Response to a Probe Meal

Participants reported to the laboratory in the morning after an overnight fast and consumed a probe meal consisting of 200 g of white beans, 18 g of wholemeal toasts, and 200 mL of peach juice (774 Kcal, 16 g of fiber). The volume of gas evacuated by anus was measured for 4 h after the probe meal, as previously described [9,23]. Briefly, gas was collected using a rectal balloon catheter (20 F Foley catheter, Bard, Barcelona, Spain) connected via a gas-tight line to a barostat, and the volume was continuously recorded. The intrarectal balloon was inflated with 5 mL of water to prevent anal gas leaks. The amounts of H_2_, CO_2_, CH_4_, O_2_, N_2_, H_2_S, and CH_4_S in the 4-h intestinal gas collection sample were measured by gas chromatography (Appendix A: analysis of intestinal gas composition).

### 2.6. Exploratory Outcomes

#### 2.6.1. Colonic Gas Content

In a subgroup of participants (*n* = 33), colonic gas content was measured by abdominal magnetic resonance imaging in the run-in phase (day 18) and in the administration phase (day 46); both occasions on the challenge diet (Figure 1; Appendix A: measurement of colonic gas volume and distribution; Appendix A: examples of regions of interest).

#### 2.6.2. Fecal Microbiota Analysis

A total of 309 fecal samples collected by participants were used for microbial community analysis. Fecal samples were collected at 5 time points throughout the study: (a) during run-in phase on the habitual diet (2 samples on days 8 and 13); (b) during run-in phase on the challenge diet (day 18); (c) in the administration phase on the habitual diet (day 41); and (d) in the administration phase on the challenge diet (day 46) (Figure 1). Participants were given standard instructions for stool sample collection: stool samples were mixed with a spatula provided to obtain a homogenous mixture, immediately frozen by the participants in their home freezers at −20 °C and later brought to the laboratory in a freezer pack, where they were stored at −80 °C until further use. Genomic DNA was extracted by mechanical process [24]. Fecal microbiota was profiled using 16S rRNA gene amplicon sequencing based on Illumina MiSeq technology. Amplicon reads were analyzed using QIIME software (1.9.1). Sequences were clustered based on the USEARCH (search and clustering) algorithm (5.2.236v) into operational taxonomic units (OTUs), taxonomically assigned with a database combining Greengenes (gg_13_8 release) and PATRIC (Pathosystems Resource Integration Center).

### 2.7. Statistical Analysis

No sample-size calculation was performed for this exploratory study. Analyzed populations for respectively the run-in and administration phases were the Full Analysis Set 1 (FAS1; *n* = 74) and FAS2 (*n* = 66). No multiplicity adjustments were performed according to the exploratory nature of the study.

#### 2.7.1. Clinical Parameters

Results are expressed as mean [95% CI]. For each item, the values of the 2 last days of each evaluation period were averaged for statistical comparisons. Comparisons were performed using Wilcoxon signed rank test. Spearman correlations were used to explore the robustness of correlations between changes in clinical parameters. Statistical tests were conducted two-sided with a significance level of 5%. All confidence intervals are presented two-sided with a confidence level of 95%.

#### 2.7.2. Gut Microbiota

Microbial ecology and statistical analyses were performed using QIIME and R software (3.4.3v). A random forest machine learning model (R package) was used to predict the evolution of clinical parameters using changes in fecal microbiota composition. Spearman correlations were used to assess correlations between changes in clinical parameters and fecal microbiota composition (Appendix A: microbiota analysis).

## 3. Results

### 3.1. Demographics and Compliance to Study Procedures

Seventy-four subjects (42 women, 32 men; 29.1 ± 7.8 years; 22.9 ± 2.3 Kg/m^2^ BMI; see Appendix A: demographics and clinical characteristics of the subjects at inclusion) were enrolled in the run-in phase, 66 fulfilled the continuation criteria and entered the administration phase, and 63 completed the study (Figure 2). Adherence to dietary instructions was high (mean per-protocol compliance >90%) and the study product was well tolerated.

### 3.2. Effect of the Flatulogenic Diet during the Run-in Phase

On their habitual diet, healthy subjects reported no significant gas-related symptoms, positive digestive well-being, and normal bowel habit with a mean of 8.7 [8.2; 9.2] daily anal gas evacuations (Table 1). The 3-day flatulogenic diet induced gas-related symptoms and impaired the sensation of digestive well-being; these subjective changes were associated with an objective increase in the number of daytime anal gas evacuations, without changes in stool frequency and consistency (Table 1).

### 3.3. Effect of the Fermented Milk Product Consumption on the Tolerance of the Flatulogenic Diet

The tolerance of the flatulogenic diet during consumption of the fermented milk product (FMP) was significantly better than during the run-in period, with lower gas-related symptoms scores (e.g., flatulence reduced by -1.7 [−1.9; −1.6]; *p* < 0.0001), smaller number of anal gas evacuations (decrease by −5.8 [−6.5; −5.1]; *p* < 0.0001), and higher digestive well-being scores (increase by +0.6 [+0.4; +0.7]; *p* < 0.05) (Figure 3). FMP consumption mitigated but did not entirely prevent the impact of the flatulogenic diet (Figure 3) as gas-related symptoms scores and the number of gas evacuations were still higher and digestive well-being worse than during the run-in period on the habitual diet. No significant differences in stool frequency and consistency were detected (changes by 0.0 [−0.1; 0.0] daily bowel movements and 0.1 [0.0; 0.1] Bristol score). The effects of the FMP on flatulence sensation correlated with the effect on the number of anal gas evacuations (Spearman rho = 0.618; *p* < 0.0001).

Mean anal gas collection during the 4 h after ingestion of the probe meal was 174 mL [163; 185] in the run-in phase and 167 mL [154; 181] in the administration phase (*p* > 0.05); no differences in gas composition were detected (Appendix A: effect of the fermented milk product on intestinal gas composition).

### 3.4. Exploratory Outcomes

#### 3.4.1. Colonic Gas Volume

The volume of gas within the colon measured by magnetic resonance imaging (MRI) was 150 mL [140; 160] in the run-in phase and 145 mL [136; 155] in the administration phase (*p* > 0.05). No differences in colonic gas distribution were detected (Appendix A: effect of the fermented milk product on intestinal gas distribution).

#### 3.4.2. Fecal Microbiota

No differences in the overall gut microbiota composition among the 5 fecal samples taken during the study were detected by weighted Unifrac distance. FMP consumption did not elicit significant changes in the fecal microbiota diversity as shown by alpha and beta-diversity (Wilcoxon signed rank tests, *p* = 0.41 for Chao1 index, *p* = 0.67 for weighted UniFrac distance) (Appendix A: evolution of microbiota alpha and beta diversity).

To predict the evolution of clinical parameters based on changes in fecal microbiota composition, we used random forest models combined with Spearman correlation. We identified associations between the effect of the intervention on specific clinical parameters and the relative abundance of some genera: the decrease in number of anal gas evacuations during FMP consumption correlated with a decrease in the relative abundance of *Mogibacterium* (rho = 0.31; *p* = 0.02) and *Parvimonas* (rho = 0.32; *p* = 0.02) and an increase in *Desulfovibrionaceae* (rho = −0.28; *p* = 0.04) (Figure 4A); the decrease in flatulence sensation was related with a decrease in the relative abundance of *Methanobrevibacter* spp. (rho = 0.39; *p* = 0.003) and *Cerasicoccaceae* (rho = 0.28; *p* = 0.039), and an increase in *Succinivibrio* (rho = −0.33; *p* = 0.014) (Figure 4B).

## 4. Discussion

Our pilot, proof-of-concept study indicates that consumption of a fermented milk product with B. *lactis* CNCM I-2494 and lactic acid bacteria may improve the tolerance of a challenge diet rich in fermentable residues in healthy subjects, suggesting that probiotics may render healthy individuals more resilient to dietary diversions.

As in previous studies [9], healthy subjects without digestive symptoms on their habitual diet became symptomatic when challenged with a flatulogenic diet. The challenge diet induced predominant symptoms that are commonly attributed to intestinal gas, such as abdominal bloating, distension, and flatulence, and had also a negative impact on digestive well-being. These sensations were associated with a substantial increase in the number of anal gas evacuations measured by an event marker. This method has been previously used with reproducible and consistent results [9,23]; furthermore, it was shown that marked evacuations exhibited a very good correlation with direct recording of anal gas outflow obtained simultaneously (R > 0.95; *p* < 0.05) [25]. The effects of the challenge diet increased over the 3-days administration period, as opposed to the stability of repeated measurements on the habitual diet in the pretreatment phase; however, the daily increments became gradually smaller conceivably tending towards a plateau with longer administration. The challenge diet did not affect normal bowel habit; a similar result was observed in a previous study showing that a 3-day high-residue diet increased the fecal output, a parameter not measured in the present study, without changes in stool frequency and consistency [26].

It has been shown that patients with functional gut disorders complaining of gas-related symptoms exhibit the same symptoms and also a number of anal gas evacuations above the normal range [9]; hence, the challenge diet in healthy subjects to some extent mimicked this clinical condition.

This experimental model of gut discomfort was used in the present study to test the effect of a fermented milk product with B. *lactis* CNCM I-2494 and lactic acid bacteria in healthy subjects. In this model, consumption of the FMP over a 4-week period had a positive impact on the 3 parameters that were affected by the challenge diet: the FMP reduced gas-related symptoms, reduced the number of anal gas evacuations, and improved digestive well-being. The FMP induced these effects without affecting normal bowel habit in healthy subjects, a response analogous to that of the challenge diet.

In previous studies, the effect of different dietary interventions on intestinal gas production was detected by changes in the number of daily anal gas evacuations measured over several days, as well as by changes in the volume of gas evacuated per anus measured during the postprandial period after a probe meal [9,23,27]. In the present study, the FMP reduced the number of anal gas evacuations, but no effect on the gas volume evacuated after the probe meal was detected, conceivably because the probe meal had a high content of residues, and the potential effect of probiotics was overflown by the abundance of fermentable substrates within the colonic biomass. Proof of a modulatory effect of a FMP with the same B. *lactis* CNCM I-2494 on intestinal gas production was provided by a previous study showing a reduction in postprandial hydrogen breath excretion after a nutrient meal containing lactulose in IBS patients [18].

No differences in the volume of intestinal gas measured by abdominal MRI were detected during the FMP administration. In a previous study, 3-day dietary interventions comparing high-fiber (35 g/d) versus low-fiber (8 g/d) diets did not result in significant changes in colonic gas volume measured by MRI during fasting [26]. Intestinal gas homeostasis is tightly regulated to maintain the volume of intraluminal gas constant under different conditions. Indeed, over 70% of the gas produced in the process of microbiota fermentation of meal residues is absorbed into the blood and eliminated by breath [28], and another part of the gas produced is disposed by gas-consuming microorganisms [5].

In a previous study, a FMP with B. *lactis* CNCM I-2494 reduced the *Prevotella/Bacteroides* ratio metabolic potential in IBS patients [18]. These results were not reproduced in the present study using a different dietary challenge in healthy subjects. However, we identified some bacterial genera that were potentially associated with the reduction in symptoms and anal gas evacuations during FMP consumption. Some of these specific genera are linked with the metabolism of hydrogen, including *Methanobrevibacter*, *Succinivibrio*, and *Desulfovibrionaceae* that might reflect a trade-off in the balance between gas-producing and gas-consuming microorganisms. For these analyses, we applied first machine learning to evaluate whether microbiota features could predict flatulence variation; then, we evaluated the most important features with Spearman correlation. We acknowledge that the microbial signal is weak and this could be related to the two sets of statistical methods used. Furthermore, our results must be considered only as hypothesis generating since no multiplicity adjustments were performed according to the exploratory nature of the study.

The reduction of symptoms and changes in the gas evacuation pattern associated to FMP administration could be not only related to microbiota metabolism of intraluminal substrates, but also to an effect of probiotics on gut sensitivity and/or on the handling of contents. A FMP with the same B. *lactis* CNCM I-2494 was previously shown to reduce visceral sensitivity to colorectal distension in a rat model [29] and to modulate the activity of brain regions that control central processing of emotion and sensation in healthy women [30]; furthermore, the same product accelerated orocecal and colonic transit in patients with IBS-C along with a reduction in symptoms and objective abdominal distension [16].

We acknowledge that in the absence of a control arm, a possible placebo effect on subjective perception cannot be excluded. However, participants were not aware of the potential effects, whether beneficial or deleterious, that the challenge diet or the FMP could produce.

## 5. Conclusions

The relevance of our study is two-fold. In the first place, the experimental model in healthy subjects presented here may be useful in the experimental setting for testing interventions aimed at improving diet-related symptoms, and this may be important because demonstrating a beneficial effect of probiotics on normal digestive function and perception in healthy subjects is a challenge. Our data may have also practical applications: high-residue diets have beneficial effects but are poorly tolerated and probiotics may enhance compliance to these healthy diets; conversely, probiotics may be indicated to reduce the side effects of dietary transgressions.

## Figures and Tables

**Figure 1 nutrients-12-00320-f001:**
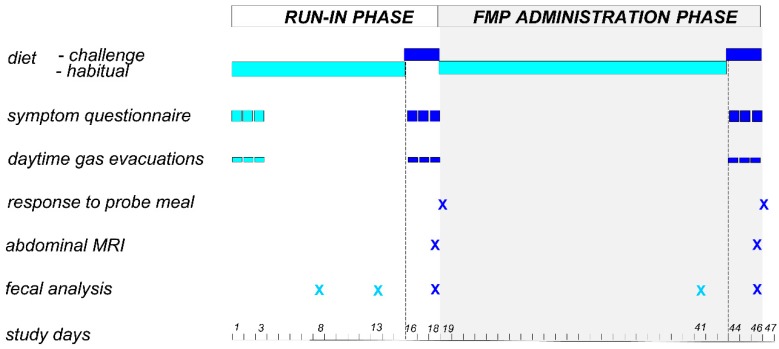
Study design. FMP: fermented milk product.

**Figure 2 nutrients-12-00320-f002:**
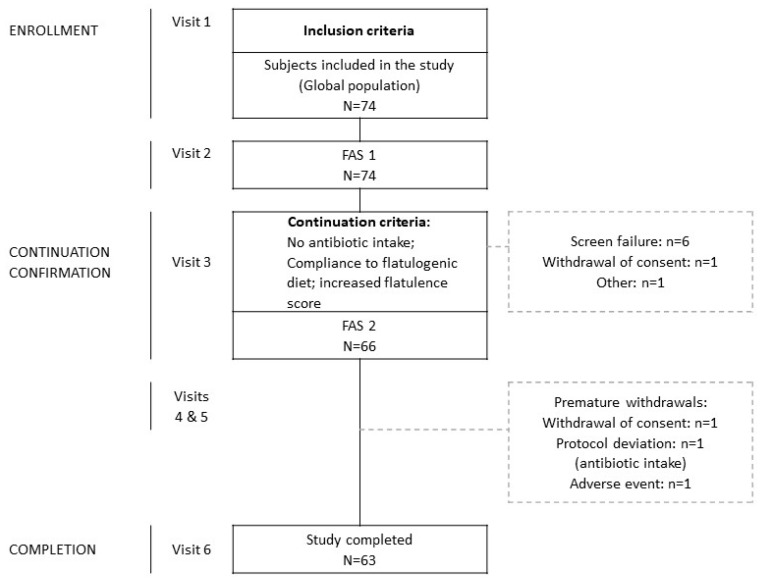
Flow-chart. FAS 1: Full Analysis Set 1; FAS 2: Full Analysis Set 2. Adverse event corresponds to urinary infection.

**Figure 3 nutrients-12-00320-f003:**
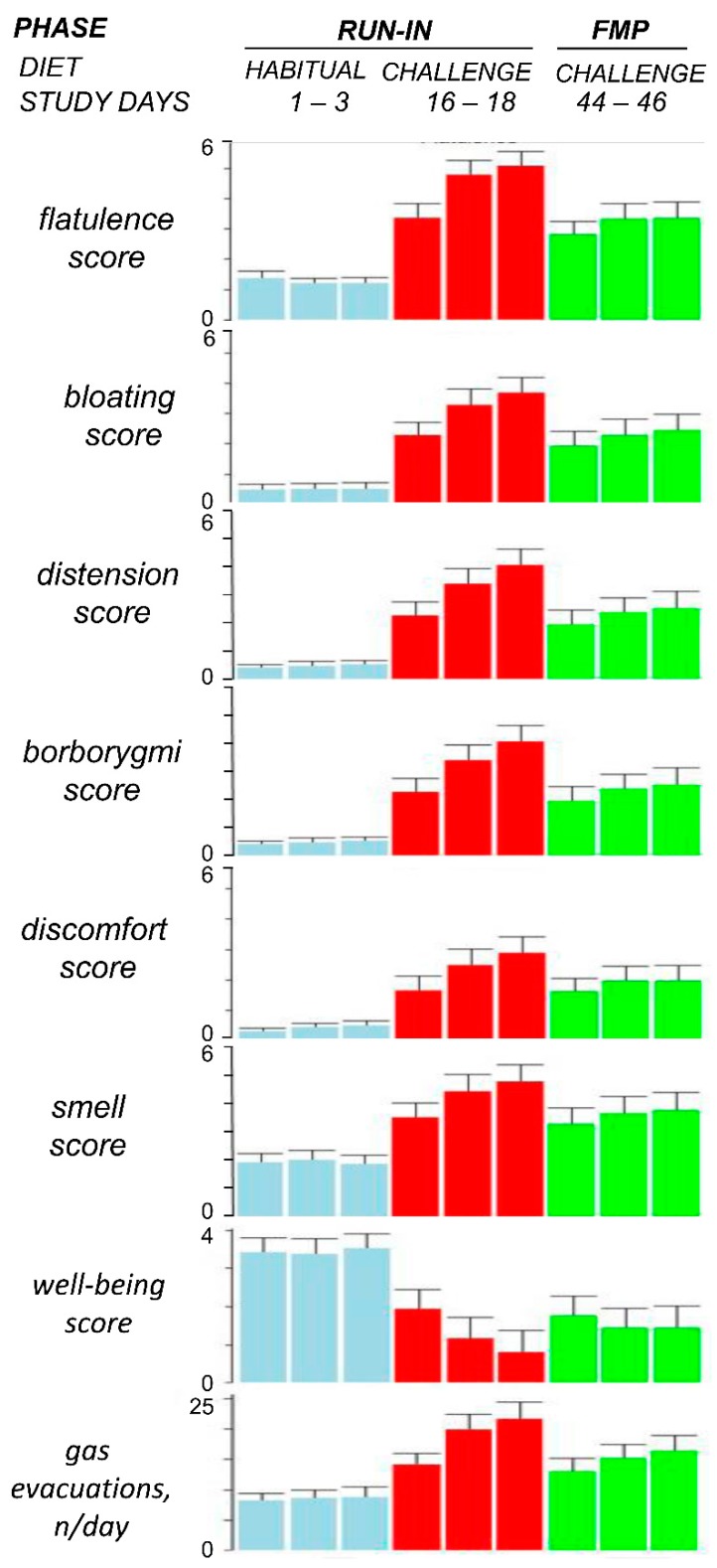
Tolerance of the flatulogenic diet. FMP: fermented milk product administration phase. To show daily variability, data of the 3 days of each evaluation period are represented; note similar values on days 2 and 3 within each period. Statistical comparisons were performed by Wilcoxon signed rank test using the average of days 2 and 3 of each period: *p* < 0.0001 for all parameters habitual diet vs. challenge diet during run-in phase (*n* = 74); *p* < 0.0001 for all parameters challenge diet during run-in phase vs. challenge diet during FMP administration phase (*n* = 66) except for abdominal discomfort/pain and digestive well-being (*p* < 0.05).

**Figure 4 nutrients-12-00320-f004:**
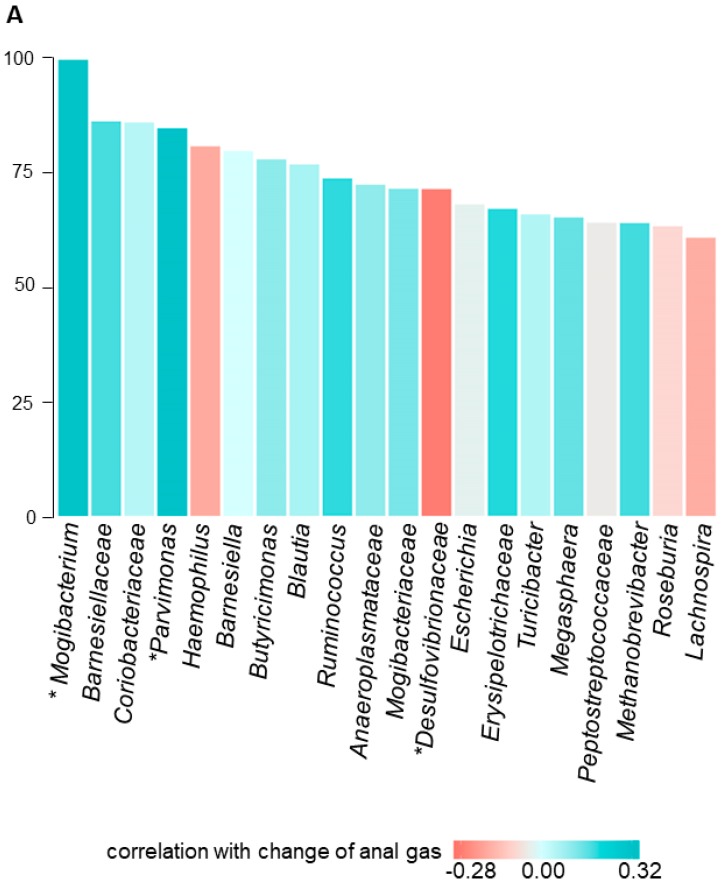
Random forest relative variable importance of most relevant microbiota genera associated with change in clinical parameters upon fermented milk product (FMP) consumption. Top 20 most important variables from random forest to predict changes between study conditions “flatulogenic diet” and “flatulogenic diet + FMP” of (**A**) anal gas (number of anal gas evacuations) and (**B**) flatulence. Strength and significance of Spearman correlations (rho coefficient) are indicated in the color shade. Red: change in microbiota genera negatively associated with the change in clinical parameter. Blue: change in microbiota genera positively associated with the change in clinical parameter. * (*p* < 0.05).

**Table 1 nutrients-12-00320-t001:** Effects of the flatulogenic diet on gas-related symptoms and frequency of anal gas evacuations.

	Habitual Diet (*n* = 74)	Flatulogenic Diet (*n* = 74)	*p* Value
Flatulence *	1.2 [1.1; 1.2]	4.9 [4.8; 5.1]	*p* < 0.0001
Abdominal discomfort/pain *	0.5 [0.4; 0.5]	2.7 [2.5; 2.9]	*p* < 0.05
Abdominal distension *	0.4 [0.4; 0.5]	3.4 [3.3; 3.6]	*p* < 0.0001
Bloating *	0.5 [0.5; 0.6]	3.5 [3.3; 3.6]	*p* < 0.0001
Borborygmi *	0.8 [0.7; 0.8]	2.8 [2.7; 3.0]	*p* < 0.0001
Odor of flatus *	1.8 [1.7; 1.9]	4.3 [4.1; 4.5]	*p* < 0.0001
Digestive well-being *	3.4 [3.3; 3.6]	1.0 [0.8; 1.2]	*p* < 0.05
Anal gas evacuations *	8.7 [8.2; 9.2]	20.7 [19.9; 21.6]	*p* < 0.0001
Bowel movements **	1.2 [1.2; 1.3]	1.4 [1.4; 1.5]	*p* > 0.05
Stool consistency **	3.7 [3.6; 3.7]	3.9 [3.8; 3.9]	*p* > 0.05

Data are means [95% confidence interval] presented on Full Analysis Set 1 (FAS1) population; comparisons by Wilcoxon signed rank test. * means over last 2 days of each evaluation period; ** means over last 3 days of each evaluation period.

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
