# Peer review of "A Fermented Milk Product with B. lactis CNCM I-2494 and Lactic Acid Bacteria Improves Gastrointestinal Comfort in Response to a Challenge Diet Rich in Fermentable Residues in Healthy Subjects"

_nutrients, 2020, doi:10.3390/nu12020320_

Round 1

Reviewer 1 Report

This study is addressing the issue of a diet rich in fermentable food residuals can induce symptoms due to gas production form colonic fermentation of the food residual. The study hypothesis is that a fermented milk product (FMP) containing multiple strains of four probiotic species, may potentially alter the colonic gut microbiota and reduce gas production from colonic fermentation of food residuals which may aid in alleviating symptoms. The study describes an open label pilot trial of the FMP in a cohort of 74 healthy adults without gastrointestinal disease. Overall the study presents low quality evidence of the effectiveness of the FMP to reduce symptoms when consuming a flatulogenic diet. However, the authors state that the study is a pilot, proof-of-concept study and acknowledge the limitations of the study.

There are a number of issues with the manuscript in its current form that the authors should address:

Results presented in the abstract are not presented in the main text.

Line 38 “experimented” should be “experienced”

The authors should provide more information on how study participants were recruited, whether the challenge diet and FMP was provide for the participants and whether the participants received any gifts or reimbursements.

More information should be given on the excluded treatment (i.e. treatments that modified gastrointestinal and central nervous system function).

The title and abstract highlights one bacteria (B. lactis) while the FMP actually contains 6 strains of 4 probiotic species. Furthermore, there is no evidence that B. lactis has any role in the observed effects of the FMP. Therefore highlighting B. lactis in the title is somewhat misleading.

How did participant homogenize stool samples at home?

Can the author provide more information on the 1 Adverse Event

Figure 3 is problematic. Statistical analysis was undertaken by the average of the scores from the last 2 days. The figure presents scores from each day individually but quotes the statistical result from the average of the last 2 days. To be transparent the authors should present the data as it was analyzed which is the average score of the last 2 days.

The authors should provide more information about the supplementary material in the main text. For example, it is not mentioned in the main text that the relative abundance of the probiotic species over the study is provide in Figure S2

Author Response

Responses to the comments of Reviewer 1

Comments and Suggestions for Authors

This study is addressing the issue of a diet rich in fermentable food residuals can induce symptoms due to gas production form colonic fermentation of the food residual. The study hypothesis is that a fermented milk product (FMP) containing multiple strains of four probiotic species, may potentially alter the colonic gut microbiota and reduce gas production from colonic fermentation of food residuals which may aid in alleviating symptoms. The study describes an open label pilot trial of the FMP in a cohort of 74 healthy adults without gastrointestinal disease. Overall the study presents low quality evidence of the effectiveness of the FMP to reduce symptoms when consuming a flatulogenic diet. However, the authors state that the study is a pilot, proof-of-concept study and acknowledge the limitations of the study.

There are a number of issues with the manuscript in its current form that the authors should address:

Results presented in the abstract are not presented in the main text.

The effects of the challenge diet are shown in Table 1. The effects of FMP had been included in the Results section, lines 278-280.

Line 38 “experimented” should be “experienced”

Corrected, thanks.

The authors should provide more information on how study participants were recruited, whether the challenge diet and FMP was provide for the participants and whether the participants received any gifts or reimbursements.

Information included in Section 2.1 Studies subjects.

More information should be given on the excluded treatment (i.e. treatments that modified gastrointestinal and central nervous system function).

Done, line 110.

The title and abstract highlights one bacteria (B. lactis) while the FMP actually contains 6 strains of 4 probiotic species. Furthermore, there is no evidence that B. lactis has any role in the observed effects of the FMP. Therefore highlighting B. lactis in the title is somewhat misleading.

Title revised as suggested.

How did participant homogenize stool samples at home?

Participants were given standard instructions for stool sample collection: stool samples were mixed with a spatula provided to obtain a homogenous mixture, line 212.

Can the author provide more information on the 1 Adverse Event

It corresponds to a urinary infection that fell into the definition of adverse event. Information included in Legend to Figure 2.

Figure 3 is problematic. Statistical analysis was undertaken by the average of the scores from the last 2 days. The figure presents scores from each day individually but quotes the statistical result from the average of the last 2 days. To be transparent the authors should present the data as it was analyzed which is the average score of the last 2 days.

The following clarification has been included in the Legend to Figure 3. “To show daily variability, Figure 3 represents data of the 3 days of each evaluation period; note similar values on days 2 and 3 within each period. Statistical comparisons were performed by Wilcoxon signed rank test using the average of days 2 and 3 of each period”.  

The authors should provide more information about the supplementary material in the main text. For example, it is not mentioned in the main text that the relative abundance of the probiotic species over the study is provide in Figure S2

All supplemental material (text, figures and tables) have been referred to in the text, as suggested.

Reviewer 2 Report

The authors focus on evaluating of sensation and number of anal gas evacuations to a challenge diet rich in fermentable residues in healthy subjects and the improvement by additional consumption of a fermented milk product with several probiotica.  Since the treatment of gastrointestinal symptoms is very complex and needs personalized nutrition, probiotic food is a very important tool for improvement.

Although these results are very contributing, there are some issues that should be taken into consideration:

Title

Probiotics should be mentioned all or not at all. Tolerance may not be the optimal word, because it suggests the context of compliance Therefore: A fermented milk product improves gastrointestinal comfort to a challenge diet…

Abstract

Line 18: ..improving the digestive comfort..

Line 19: ..66? healthy subjects..

Line 20: ..containing..: again autors should specify all of no probiotic also in line 29.

Line 21: ..intestinal gas-related symptoms or digestive sensations like in 2.5

Introduction

A small paragraph on the importance of the in principal positive role of human intestinal gases, like their role as gasotransmitters in anti-inflammatory, -oxidative and neuroprotective context is missing.

 Subjects and Methods

Were there men and women in the investigated group? Give more details to the food habits of the subjects. Did they have a constant omnivorous or vegetarian or…diet for at least 6 months? What about allergies or food intolerances? Specify in more detail the used questionnaires and their standardization quality. It is important to give some information to the time needed for data assessment. Was there given any standardized instruction to the participants? There should be explained, why you defined the number of days in the run-in phase. The detailed information on qualitative and quantitative nutritive assessment is missing. Details on the background information to the challenge diet are missing. What was the nutrient and fiber content and what are the considerations in comparison to real life nutrition.

Results

Data and SD (age, gender, BMI) of participants is missing

 Discussion

The authors should discuss the importance of this study outcomes to daily food intake and/or nutraceutical applications.

Author Response

Responses to the comments of Reviewer 2

The authors focus on evaluating of sensation and number of anal gas evacuations to a challenge diet rich in fermentable residues in healthy subjects and the improvement by additional consumption of a fermented milk product with several probiotica.  Since the treatment of gastrointestinal symptoms is very complex and needs personalized nutrition, probiotic food is a very important tool for improvement.

Although these results are very contributing, there are some issues that should be taken into consideration:

Title

Probiotics should be mentioned all or not at all.

Tolerance may not be the optimal word, because it suggests the context of compliance Therefore: A fermented milk product improves gastrointestinal comfort to a challenge diet…

Title revised as suggested.

Abstract

Line 18: ..improving the digestive comfort..

Line 19: ..66? healthy subjects..

Line 20: ..containing..: again autors should specify all of no probiotic also in line 29.

Line 21: ..intestinal gas-related symptoms or digestive sensations like in 2.5

Text revised following up the comments above.

Introduction

A small paragraph on the importance of the in principal positive role of human intestinal gases, like their role as gasotransmitters in anti-inflammatory, -oxidative and neuroprotective context is missing.

Done, lines 73-76.

Subjects and Methods

Were there men and women in the investigated group?

Yes, information included in lines 100 and 151.

Give more details to the food habits of the subjects. Did they have a constant omnivorous or vegetarian or…diet for at least 6 months? What about allergies or food intolerances?

All participants followed a non-restrictive, omnivorous diet without any change in dietary habits in the previous 4 weeks; allergies and food intolerances were excluded. Information included in Section 2.1.

Specify in more detail the used questionnaires and their standardization quality. It is important to give some information to the time needed for data assessment. Was there given any standardized instruction to the participants?

Information included in Section 2.1.

There should be explained, why you defined the number of days in the run-in phase.

A 15-day period in the run-in phase was allowed before the first flatulogenic diet to avoid potential carry-over effects of probiotic consumption in the habitual diet of the participants prior to their inclusion in the study. Information included in lines 138-140.

The detailed information on qualitative and quantitative nutritive assessment is missing. Details on the background information to the challenge diet are missing. What was the nutrient and fiber content and what are the considerations in comparison to real life nutrition.

Information included in Section 2.3.

Results

Data and SD (age, gender, BMI) of participants is missing

Data included in line 251.

Discussion

The authors should discuss the importance of this study outcomes to daily food intake and/or nutraceutical applications.

Done as suggested.

Round 2

Reviewer 1 Report

The authors have addressed all comments

Reviewer 2 Report

The authors submitted the corrected file according to suggestions.